# Adaptation of hepatitis C virus to interferon lambda polymorphism across multiple viral genotypes

Nimisha Chaturvedi[1,2]*, Evguenia S Svarovskaia[3], Hongmei Mo[3], Anu O Osinusi[3], Diana M Brainard[3], G Mani Subramanian[3], John G McHutchison[3], Stefan Zeuzem[4], Jacques Fellay[1,2,5]*

[1]School of Life Sciences, École Polytechnique Fédérale de Lausanne, Lausanne, Switzerland; [2]Swiss Institute of Bioinformatics, Lausanne, Switzerland; [3]Gilead Sciences Inc, Foster City, United States; [4]Goethe University Hospital, Frankfurt, Germany; [5]Precision Medicine Unit, Lausanne University Hospital, Lausanne, Switzerland

**Abstract** Genetic polymorphism in the interferon lambda (IFN-λ) region is associated with spontaneous clearance of hepatitis C virus (HCV) infection and response to interferon-based treatment. Here, we evaluate associations between IFN-λ polymorphism and HCV variation in 8729 patients (Europeans 77%, Asians 13%, Africans 8%) infected with various viral genotypes, predominantly 1a (41%), 1b (22%) and 3a (21%). We searched for associations between rs12979860 genotype and variants in the NS3, NS4A, NS5A and NS5B HCV proteins. We report multiple associations in all tested proteins, including in the interferon-sensitivity determining region of NS5A. We also assessed the combined impact of human and HCV variation on pretreatment viral load and report amino acids associated with both IFN-λ polymorphism and HCV load across multiple viral genotypes. By demonstrating that IFN-λ variation leaves a large footprint on the viral proteome, we provide evidence of pervasive viral adaptation to innate immune pressure during chronic HCV infection.

DOI: https://doi.org/10.7554/eLife.42542.001

*For correspondence:
chaturvedi.nimisha20@gmail.com (NC);
jacques.fellay@epfl.ch (JF)

## Introduction

Infection with hepatitis C virus (HCV), a positive strand RNA virus of the Flaviviridae family, represents a major health problem, with an estimated 71 million chronically infected patients worldwide (**WHO, 2017**). In the absence of treatment, 15–30% of individuals with chronic HCV infection develop serious complications including cirrhosis, hepatocellular carcinoma and liver failure (**Shepard et al., 2005**; **Alter and Seeff, 2000**; **Li et al., 2015**; **Drummer, 2014**).

Seven major genotypes of HCV have been described, further divided into several subtypes (**Simmonds, 2004**; **Smith et al., 2014**). Moreover, within each infected individual, multiple distinct HCV variants co-exist as quasipecies (**Farci et al., 2000**). Inter-host and intra-host HCV evolution is shaped by multiple forces, including human immune pressure (**Merani et al., 2011**). To investigate the complex interactions between host and pathogen at the level of genetic variation, we proposed a genome-to-genome approach that allows the joint analysis of host and pathogen genomic data (**Bartha et al., 2013**). Using an unbiased association study framework, a genome-to-genome analysis aims at identifying the escape mutations that accumulate in the pathogen genome in response to host genetic variants. **Ansari et al. (2017)** used this approach to analyze a cohort of individuals of white ancestry predominantly infected with genotype 3a HCV; they identified associations between viral variants and human polymorphisms in the interferon lambda (IFN-λ) and HLA regions,

demonstrating an impact of both innate and acquired immunity on HCV sequence variation during chronic infection.

The IFN-λ association is of particular interest considering the known impact of this polymorphic region on spontaneous clearance of HCV and on response to interferon-based treatment (*Ge et al., 2009*; *Rauch et al., 2010*; *Thomas et al., 2009*; *Tanaka et al., 2009*). The rs12979860 variant, which is located 3 kb upstream of *IL28B* (encoding IFN-λ3) and lies within intron 1 of *IFNL4*, showed the strongest correlation with treatment-induced clearance of infection in the first report (*Ge et al., 2009*). More recent studies have shown that rs12979860 is in fact a marker for a dinucleotide insertion/deletion polymorphism, *IFNL4* rs368234815 [ΔG > TT], which causes a frameshift that abrogates IFN-λ4 protein production (*Prokunina-Olsson et al., 2013*). The two variants (rs12979860 and rs368234815) are in strong linkage disequilibrium in European and Asian populations (r2 = 0.98 in CEU and 1.00 in CHB and JPT): the rs12979860 C allele, associated with a higher rate of spontaneous HCV clearance and better response to interferon-based treatment, is found on the same haplotype as the rs368234815 TT allele and is thus tagging the absence of IFN-λ4 protein.

Here, we aim at characterizing the importance of innate immune response in modulating chronic HCV infection by describing the footprint of *IFNL4* variation in the viral proteome. Using samples and data from a heterogeneous group of 8,729 HCV-infected individuals in a cross-sectional study design, we genotyped the single nucleotide polymorphism (SNP) rs12979860 and obtained partial sequences of the HCV genome (*NS3*, *NS4A*, *NS5A* and *NS5B* genes). We tested for associations between rs12979860, HCV amino acid variants and pre-treatment viral load. We show that the presence or absence of the IFN-λ4 protein has a pervasive impact on HCV, by describing multiple associations between host and pathogen variants in subgroups defined by viral genotype or human ancestry. We also present association analyses of human and viral variants with HCV viral load, which allows for a better understanding of the connections between genomic variation, biological mechanisms and clinical outcomes.

## Results

### Host and pathogen data

We obtained paired human and viral genetic data for 8,729 HCV-infected patients participating in various clinical trials of anti-HCV drugs. The samples were heterogeneous in terms of self-reported ancestry (85% Europeans, 13% Asians and 2% Africans) and HCV genotypes, with a majority of HCV genotype 1a, 2a and 3a (*Table 1*). We genotyped the human SNP rs12979860 and performed deep sequencing of the coding regions of the HCV non-structural proteins NS3, NS4A, NS5A and NS5B (*Bartenschlager et al., 2004*). A binary variable was generated for each alternate amino acid, indicating the presence or absence of that allele in a given sample (N = 10,681). For the analysis, we used only amino acids that were present in at least 0.3% of the samples (N = 4,022).

**Table 1.** Characteristics of study participants, by HCV genotype group.

| HCV genotype | All | 1a | 1b | 2a | 2b | 3a | 4a | Others |
|---|---|---|---|---|---|---|---|---|
| *N* | *8729* | *3548 (41)* | *1924 (22)* | *304 (3)* | *472 (5)* | *1839 (21)* | *193 (2)* | *449 (5)* |
| Europeans | 6704 (77) | 2987 (84) | 1133 (59) | 100 (33) | 421 (89) | 1635 (89) | 178 (92) | 250 (56) |
| Asians | 1103 (13) | 59 (2) | 577 (30) | 197 (65) | 15 (3) | 111 (6) | 2 (1) | 142 (32) |
| Africans | 723 (8) | 421 (12) | 192 (10) | 7 (2) | 25 (5) | 19 (1) | 8 (4) | 51 (11) |
| Others | 199 (2) | 81 (2) | 22 (1) | 0 (0) | 11 (2) | 74 (4) | 5 (3) | 6 (1) |
| Cirrhosis | 2410 (28) | 978 (28) | 536 (28) | 35 (12) | 77 (16) | 629 (34) | 60 (31) | 95 (21) |
| Male sex | 5605 (64) | 2434 (69) | 1096 (57) | 141 (46) | 301 (64) | 1230 (67) | 143 (74) | 260 (58) |
| SVR | 7702 (88) | 3240 (91) | 1773 (92) | 273 (90) | 426 (90) | 1452 (79) | 153 (79) | 385 (86) |

Data are indicated as number (percent); SVR: sustained virological response after treatment.

DOI: https://doi.org/10.7554/eLife.42542.002

## Associations between IFN-λ polymorphism and HCV amino acids

We performed a separate analysis for each HCV genotype, using an additive logistic model with binary amino acid variables as traits of interest. To control for population stratification, we added host and viral covariates in the model and to control for multiple testing we used a Bonferroni threshold of $4.7 \times 10^{-6}$, which was calculated based on the number of tests performed (more information in the Materials and methods section). We restricted the analysis to genotypes 1a, 1b, 2a, 2b, 3a and 4a, which were present in at least 100 participants.

We observed highly significant associations between rs12979860 and HCV amino acid variables for each HCV genotype that we examined (*Figure 1*, *Table 2*). The highest number of significant associations was detected in the largest group of patients, infected with genotype 1a, most likely reflecting an effect of sample size on statistical power. Most associations were specific to a single viral genotype; however, some associations were significant across genotypes. As an example, two strong associations were observed between rs12979860 and amino acid variables at position 2576 in viral protein NS5B, with the T allele associating with proline in genotypes 1a ($p=1.5\times10^{-10}$), 2b ($p=5.4\times10^{-15}$), 3a ($p=8.3\times10^{-12}$) and 4a ($p=1.2\times10^{-7}$), and the C allele associating with alanine in genotypes 1a ($p=1.2\times10^{-11}$), 2a ($p=3.8\times10^{-6}$), 2b ($p=4.02\times10^{-8}$) and 3a ($p=1.04\times10^{-14}$).

In patients infected with genotype 3a, we replicated the previously reported associations (*Ansari et al., 2017*) between *IFNL4* variation and valine at position 2570 in NS5B ($p=5.5\times10^{-20}$), histidine at position 2991 in NS5B ($p=4.6\times10^{-12}$) and asparagine at position 2414 in NS5A ($p=2.4\times10^{-7}$). We also observed novel associations with alanine ($p=2.6\times10^{-7}$) and threonine ($p=8.8\times10^{-15}$) at position 2570 in NS5B, as well as with glycine ($p=5.2\times10^{-7}$) and serine ($p=1.04\times10^{-11}$) at position 2414 in NS5A. All these associations were only detected in the 3a subgroup. In concordance with a previous study (*Peiffer et al., 2016*), we also observed a significant association with histidine at position 2065 of NS5A in patients infected with HCV genotypes 1a ($p=9.8\times10^{-7}$) and 1b ($p=1.3\times10^{-7}$).

We also observed multiple significant associations in the interferon-sensitivity determining region (ISDR, amino acid positions 2209 to 2248 in NS5A) in patients infected with genotype 1b, the strongest one being with the presence of leucine at position 2224 ($p=1.5\times10^{-12}$). For genotype 1a, we observed a single significant association in the ISDR region with the presence of leucine at position 2211 ($p=2.8\times10^{-6}$).

To check whether the association of *IFNL4* genotype with HCV amino acid variables could be dependent of the effect of *IFNL4* genotype on viral replication rates, we also compared the results from two sets of logistic regression models: one that does and one that does not include HCV viral load as an additional covariate. We did not observe any significant difference in the results of the two models (*Figure 1—figure supplement 1*).

## Viral load association analyses

To further understand the clinical implications of viral mutations associated with IFN-λ polymorphism, we searched for associations between rs12979860, HCV amino acid variants and viral load. For this, we first searched for associations between rs12979860 and Box-Cox transformed pre-treatment HCV viral load, in subgroups defined by HCV genotypes. Pre-treatment viral load was found to be significantly associated ($p<0.05$) with rs12979860 for all HCV genotypes, with the rs12979860 T allele consistently associated with lower viral load (*Figure 1—figure supplement 2*). The strength of the association p-values varied between genotypes due to sample size, but the effect size associated with the T allele was comparable across genotype groups.

We then searched for associations between viral load and HCV amino acid variables. These analyses identified significant associations in all viral genotype groups except 4a (*Figure 2*). Amongst the viral amino acids that associated with viral load, a number also associated with rs12979860 genotype (genotype 1a, 9 of 18 amino acids; 1b, 5 of 17 amino acids; 2a, 0 of 2 amino acids; 2b, 0 of 6 amino acids; 3a, 2 of 3 amino acids). As an example of such a complex association pattern, we looked at position 2224 of NS5A (in the ISDR) in genotype 1b. Mean viral load was higher in patients infected with a virus harboring a leucine in comparison to the most common amino acid alanine (t-test p-value: $5.6 \times10^{-9}$, with $H_{alternative} = \mu_{vl}^{L} - \mu_{vl}^{A} > 0$) (*Figure 3A*). This was true for both CC and non CC genotypes of SNP rs12979860 (t-test p-value: $6.2 \times10^{-6}$ for CC,L vs. CC,non-L; t-test p-value: 4.1

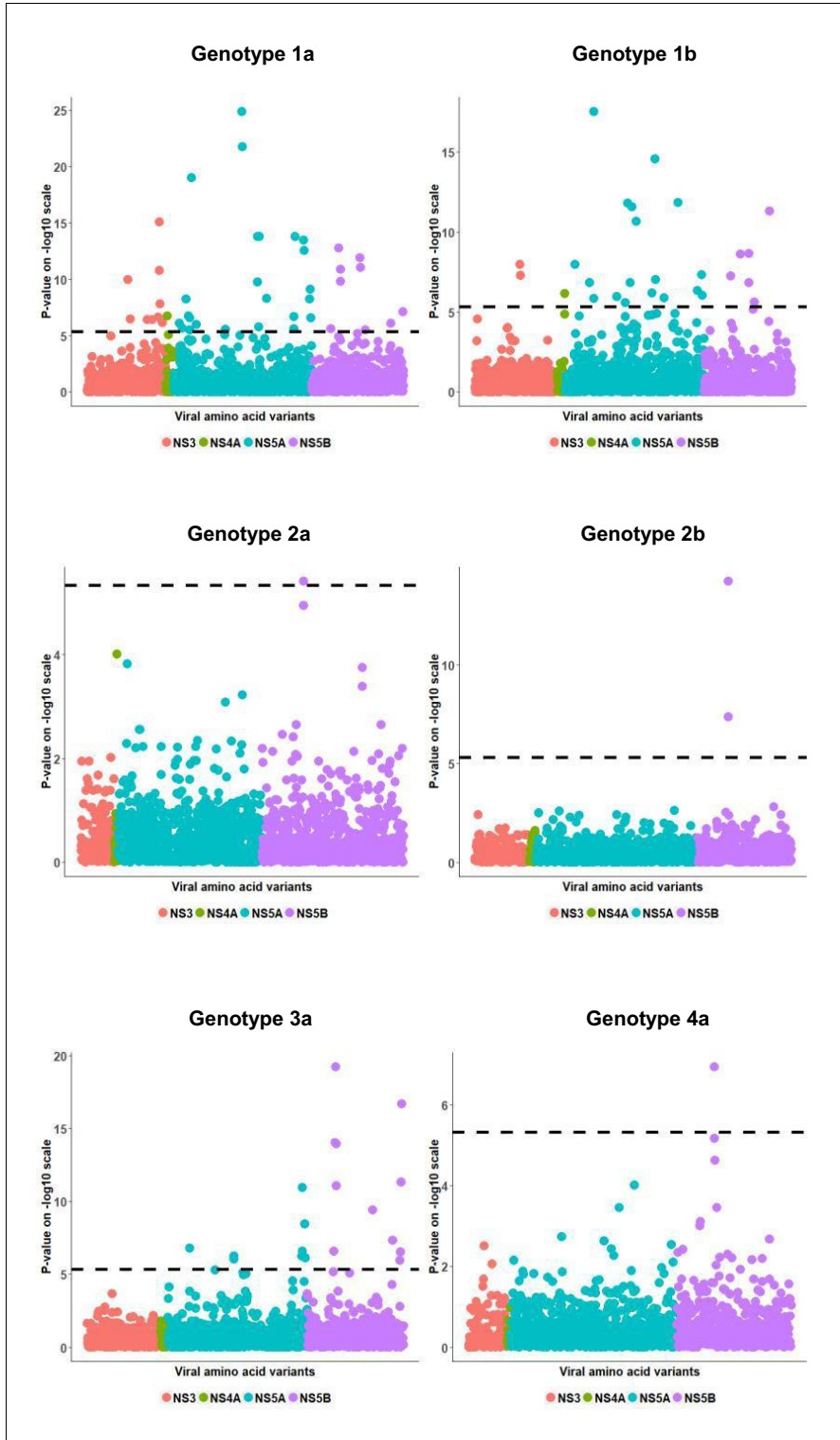

**Figure 1.** Per genotype integrated association analysis results. Manhattan plot for associations between human SNP rs12979860 and HCV amino acid variants. The dotted line shows the Bonferroni-corrected significance threshold.

DOI: https://doi.org/10.7554/eLife.42542.003

The following figure supplements are available for figure 1:

**Figure supplement 1.** Per genotype integrated association analysis results corrected for HCV viral load.

*Figure 1 continued on next page*

*Figure 1 continued*

DOI: https://doi.org/10.7554/eLife.42542.004

**Figure supplement 2.** Boxplot of transformed viral load stratified by rs12979860 genotypes (CC, CT, TT).

DOI: https://doi.org/10.7554/eLife.42542.005

x10$^{-2}$ for CT,L vs. CT,non-L), indicating a possible impact of that leucine residue on viral replication (*Figure 3B*).

We also replicated the previously shown (*Ansari et al., 2017*) association between viral load and the change from a serine to an asparagine at position 2414 in NS5A protein (p=4.5×10$^{-7}$) in genotype 3a and observed a lower mean viral load for patients with non-CC genotype and presence of serine at position 2414 (*Figure 3—figure supplement 1*).

To further understand these associations, we performed a residual regression analysis. We searched for associations between the amino acid variables and viral load residuals, obtained after regressing the transformed viral load on rs12979860. The objective of this analysis was to identify amino acids associated with changes in viral load that cannot be entirely explained by rs12979860 genotype. We observed multiple significantly associated amino acids with residual viral load across genotypes (*Figure 3—figure supplement 2*). A total of 7 amino acids in genotype 1a (*supplementary file 1*) and six amino acids in genotype 1b (*supplementary file 2*) associated with rs12979860 genotype, viral load and viral load residuals, including again leucine at position 2224 of NS5A in genotype 1b (p$_{residual}$ = 4.9×10$^{-8}$).

## Ancestry-specific sub-analyses

We also ran association analyses between IFN-λ variations and the variations in the HCV genome in subgroups defined by self-reported ancestry: European, Asian, and African. The association results are broadly similar to per genotype analysis and are presented in *supplementary file 3*.

We further dissected the association signals within the largest ancestry group, Europeans, by running a per genotype analysis within this sample (*Figure 3—figure supplement 3*). The strongest association was observed with the presence of isoleucine at position 2252 of viral protein NS5A in patients infected with HCV genotype 1a (p=1.2×10$^{-24}$). All the significant results from this study are presented in *supplementary file 4*.

Results of the ancestry-specific sub-analyses of associations with HCV viral load are comparable to the results obtained in the whole study population and are presented in *Figure 3—figure supplement 4*, *Figure 3—figure supplement 5* and *supplementary file 5*.

## Discussion

We used an integrated association analysis approach to explore the impact of human genetic variation in the IFN-λ region on part of the HCV proteome during chronic infection. Our results reveal a strong footprint of innate immune pressure on the non-structural regions of the HCV genome and provide strong evidence for pervasive HCV adaptation to innate immunity. We performed analyses in different sub-groups, which showed an impact of *IFNL4* variation on HCV across genotypes and ancestry categories. Finally, we report viral amino acids significantly associated with both *IFNL4* variation and HCV viral load, indicating that some of the HCV clinical and biological outcomes could be explained by traceable host–pathogen interactions.

Because we genotyped the human SNP rs12979860, a reliable marker for the dinucleotide insertion/deletion polymorphism rs368234815, our analyses exclusively focus on the effects of the presence or absence of the IFN-λ4 protein on HCV amino acids and viral load. Therefore, one clear limitation of our study is the impossibility to distinguish between the two haplotypes encoding the IFN-λ4 P70 and S70 isoforms, which have been shown to have distinctive influences on HCV pathogenesis (*Ansari, 2018*).

Our analysis detected multiple associations in all tested proteins, including NS5A. This protein is required for HCV RNA replication and virus assembly and has been shown to associate with interferon signaling and hepatocarcinogenesis (*Nakamoto et al., 2014*). Previous studies have also shown strong associations between variants in the ISDR of NS5A and HCV viral load as well as response to IFN-based therapy (*Enomoto et al., 1995*; *Frangeul et al., 1998*). Some of the

**Table 2.** Genome-to-genome analysis results per genotype.

The table shows significant p-values ($<4.7 \times 10^{-6}$), NA representing non-significant associations. We also give odds ratio (OR) and 97% confidence interval for each significant association.

| HCV gene | Position (amino acid) | Genotype 1a N = 3548 | Genotype 1b N = 1924 | Genotype 2a N = 304 | Genotype 2b N = 472 | Genotype 3a N = 1839 | Genotype 4a N = 193 |
|---|---|---|---|---|---|---|---|
| NS3 | 1332(A) | 1.02e-10 (OR 1.06; 1.04–1.08) | NA | NA | NA | NA | NA |
| NS3 | 1355(I) | 3.14e-07 (OR 1.1; 1.06–1.14) | NA | NA | NA | NA | NA |
| NS3 | 1370(I) | NA | 1.09e-08 (OR 0.83; 0.78–0.88) | NA | NA | NA | NA |
| NS3 | 1370(T) | NA | 4.87e-08 (OR 1.2; 1.12–1.28) | NA | NA | NA | NA |
| NS3 | 1473(D) | 3.82e-07 (OR 1.03 1.02–1.04) | NA | NA | NA | NA | NA |
| NS3 | 1516(I) | 3.51e-07 (OR 1.06; 1.04–1.09) | NA | NA | NA | NA | NA |
| NS3 | 1598(R) | 2.26e-07 (OR 1.04; 1.02–1.05) | NA | NA | NA | NA | NA |
| NS3 | 1612(I) | 7.88e-16 (OR 0.86; 0.83–0.89) | NA | NA | NA | NA | NA |
| NS3 | 1612(N) | 1.54e-11 (OR 1.09; 1.06–1.11) | NA | NA | NA | NA | NA |
| NS3 | 1612(T) | 1.54e-08 (OR 1.11; 1.07–1.15) | NA | NA | NA | NA | NA |
| NS3 | 1635(I) | 7e-07 (OR 1.1; 1.06–1.14) | NA | NA | NA | NA | NA |
| NS4A | 1671(T) | 1.83e-07 (OR 1.03; 1.02–1.04) | NA | NA | NA | NA | NA |
| NS4A | 1703(R) | NA | 6.94e-07 (OR 1.19; 1.11–1.27) | NA | NA | NA | NA |
| NS5A | 1996(R) | 7.87e-07 (OR 1.01; 1.01–1.02) | NA | NA | NA | NA | NA |
| NS5A | 2009(F) | NA | 1.04e-08 (OR 1.11; 1.07–1.15) | NA | NA | NA | NA |
| NS5A | 2009(I) | 2.01e-06 (OR 1.02; 1.01–1.02) | NA | NA | NA | NA | NA |
| NS5A | 2024(V) | 5.81e-09 (OR 1.04; 1.03–1.05) | NA | NA | NA | NA | NA |
| NS5A | 2034(D) | 1.75e-07 (OR 1.03; 1.02–1.04) | NA | NA | NA | NA | NA |
| NS5A | 2034(T) | NA | NA | NA | NA | 1.61e-07 (OR 0.91; 0.87–0.94) | NA |
| NS5A | 2040(K) | 3.05e-06 (OR 0.98; 0.97–0.99) | NA | NA | NA | NA | NA |
| NS5A | 2040(R) | 2.54e-07 (OR 1.03; 1.02–1.04) | NA | NA | NA | NA | NA |
| NS5A | 2047(A) | 9.8e-20 (OR 1.07; 1.06–1.09) | NA | NA | NA | NA | NA |
| NS5A | 2065(H) | 9.81e-07 (OR 1.01; 1.01–1.02) | 1.38e-07 (OR 1.06; 1.04–1.09) | NA | NA | NA | NA |
| NS5A | 2080(K) | NA | 2.9e-18 (OR 1.12; 1.09–1.14) | NA | NA | NA | NA |
| NS5A | 2080(R) | NA | 1.39e-06 (OR 0.95; 0.93–0.97) | NA | NA | NA | NA |

*Table 2 continued on next page*

Table 2 continued

| HCV gene | Position (amino acid) | Genotype 1a N = 3548 | Genotype 1b N = 1924 | Genotype 2a N = 304 | Genotype 2b N = 472 | Genotype 3a N = 1839 | Genotype 4a N = 193 |
|---|---|---|---|---|---|---|---|
| NS5A | 2187(R) | NA | 1.07e-06 (OR 1.07; 1.04–1.09) | NA | NA | NA | NA |
| NS5A | 2211(L) | 2.84e-06 (OR 0.99; 0.98–0.99) | NA | NA | NA | NA | NA |
| NS5A | 2220(R) | NA | 2.65e-06 (OR 1.03; 1.02–1.04) | NA | NA | NA | NA |
| NS5A | 2224(L) | NA | 1.6e-12 (OR 1.05; 1.04–1.07) | NA | NA | NA | NA |
| NS5A | 2234(W) | NA | 1.46e-07 (OR 1.06; 1.03–1.08) | NA | NA | NA | NA |
| NS5A | 2237(K) | NA | 2.6e-12 (OR 1.06; 1.04–1.08) | NA | NA | NA | NA |
| NS5A | 2251(I) | NA | 2.05e-11 (OR 1.07; 1.05–1.09) | NA | NA | NA | NA |
| NS5A | 2252(I) | 1.29e-25 (OR 1.12; 1.1–1.15) | NA | NA | NA | 8.68e-07 (OR 1.05; 1.03–1.07) | NA |
| NS5A | 2252(V) | 1.72e-22 (OR 0.89; 0.87–0.91) | NA | NA | NA | 5.5e-07 (OR 0.95; 0.92–0.97) | NA |
| NS5A | 2287(I) | 1.54e-14 (OR 1.09; 1.07–1.12) | 6.24e-07 (OR 1.08; 1.05–1.11) | NA | NA | NA | NA |
| NS5A | 2287(V) | 1.82e-10 (OR 0.92; 0.90–0.95) | NA | NA | NA | NA | NA |
| NS5A | 2298(I) | 1.56e-06 (OR 1.05; 1.03–1.08) | NA | NA | NA | NA | NA |
| NS5A | 2298(V) | 1.66e-14 (OR 0.92; 0.90–0.94) | NA | NA | NA | NA | NA |
| NS5A | 2300(P) | NA | 2.7e-15 (OR 1.12; 1.09–1.15) | NA | NA | NA | NA |
| NS5A | 2300(S) | NA | 9.41e-08 (OR 0.94; 0.91–0.96) | NA | NA | NA | NA |
| NS5A | 2320(Q) | 5.01e-09 (OR 1.08; 1.05–1.11) | NA | NA | NA | NA | NA |
| NS5A | 2330(R) | NA | 1.26e-06 (OR 1.03; 1.02–1.04) | NA | NA | NA | NA |
| NS5A | 2360(A) | NA | 1.46e-12 (OR 1.12; 1.09–1.16) | NA | NA | NA | NA |
| NS5A | 2371(S) | 2.03e-07 (OR 1.03; 1.02–1.04) | NA | NA | NA | NA | NA |
| NS5A | 2372(A) | 2.44e-06 (OR 0.96; 0.94–0.97) | NA | NA | NA | NA | NA |
| NS5A | 2372(S) | 1.63e-14 (OR 1.06; 1.04–1.07) | NA | NA | NA | NA | NA |
| NS5A | 2385(C) | 3.24e-14 (OR 1.09; 1.07–1.11) | 4.35e-07 (OR 1.04; 1.03–1.06) | NA | NA | NA | NA |
| NS5A | 2385(Y) | 2.7e-13 (OR 0.93; 0.91–0.94) | NA | NA | NA | NA | NA |
| NS5A | 2411(G) | NA | 4.61e-08 (OR 1.11; 1.07–1.15) | NA | NA | NA | NA |
| NS5A | 2411(S) | NA | 9.02e-07 (OR 0.92; 0.89–0.95) | NA | NA | NA | NA |
| NS5A | 2412(K) | 5.74e-09 (OR 1.03; 1.02–1.05) | NA | NA | NA | NA | NA |

Table 2 continued on next page

*Table 2 continued*

| HCV gene | Position (amino acid) | Genotype 1a N = 3548 | Genotype 1b N = 1924 | Genotype 2a N = 304 | Genotype 2b N = 472 | Genotype 3a N = 1839 | Genotype 4a N = 193 |
|---|---|---|---|---|---|---|---|
| NS5A | 2412(T) | 7.87e-10 (OR 0.93; 0.91–0.95) | NA | NA | NA | NA | NA |
| NS5A | 2414(D) | 2.43e-07 (OR 0.97; 0.96–0.98) | NA | NA | NA | NA | NA |
| NS5A | 2416(G) | NA | NA | NA | NA | 5.21e-07 (OR 1.06; 1.04–1.09) | NA |
| NS5A | 2416(N) | NA | NA | NA | NA | 2.5e-07 (OR 1.09; 1.05–1.12) | NA |
| NS5A | 2416(S) | NA | NA | NA | NA | 1.04e-11 (OR 0.89; 0.86–0.92) | NA |
| NS5A | 2420(N) | NA | NA | NA | NA | 3.39e-09 (OR 1.08; 1.05–1.11) | NA |
| NS5A | 2420(S) | NA | NA | NA | NA | 7.1e-07 (OR 0.95; 0.93–0.97) | NA |
| NS5B | 2510(N) | 2.25e-06 (OR 1.02; 1.01–1.03) | NA | NA | NA | NA | NA |
| NS5B | 2567(I) | 1.73e-13 (OR 1.02; 1.02–1.03) | 5.73e-08 (OR 1.07; 1.04–1.09) | NA | NA | NA | NA |
| NS5B | 2570(A) | NA | NA | NA | NA | 2.63e-07 (OR 1.11; 1.06–1.15) | NA |
| NS5B | 2570(T) | NA | NA | NA | NA | 8.87e-15 (OR 1.11; 1.08–1.14) | NA |
| NS5B | 2570(V) | NA | NA | NA | NA | 5.57e-20 (OR 0.84; 0.81–0.87) | NA |
| NS5B | 2576(A) | 1.21e-11 (OR 1.02; 1.01–1.02) | NA | 3.84e-06 (OR 1.27; 1.15–1.4) | 4.02e-08 (OR 1.2; 1.13–1.28) | 1.04e-14 OR 1.07; 1.05–1.08) | NA |
| NS5B | 2576(P) | 1.53e-10 (OR 0.98; 0.98–0.99) | NA | NA | 5.41e-15 (OR 0.77; 0.72–0.82) | 8.39e-12 (OR 0.95; 0.94–0.96) | 1.13e-07 (OR 0.83; 0.77–0.88) |
| NS5B | 2633(S) | NA | 2.33e-09 (OR 1.08; 1.06–1.11) | NA | NA | NA | NA |
| NS5B | 2729(Q) | 1.19e-12 (OR 0.91; 0.89–0.94) | 1.38e-07 (OR 0.94; 0.92–0.96) | NA | NA | NA | NA |
| NS5B | 2729(R) | 9.13e-12 (OR 1.09; 1.06–1.12) | 2.22e-09 (OR 1.08; 1.05–1.11) | NA | NA | NA | NA |
| NS5B | 2755(N) | 2.98e-06 (OR 1.04; 1.02–1.06) | NA | NA | NA | NA | NA |
| NS5B | 2758(A) | NA | 2.3e-06 (OR 1.05; 1.03–1.07) | NA | NA | NA | NA |
| NS5B | 2794(Q) | NA | NA | NA | NA | 3.56e-10 (OR 1.08; 1.05–1.1) | NA |
| NS5B | 2860(G) | NA | 4.63e-12 (OR 1.07; 1.05–1.09) | NA | NA | NA | NA |
| NS5B | 2937(K) | 8.23e-07 (OR 0.95; 0.93–0.97) | NA | NA | NA | NA | NA |
| NS5B | 2937(R) | NA | NA | NA | NA | 4.4e-08 (OR 1.08; 1.05–1.11) | NA |
| NS5B | 2986(H) | NA | NA | NA | NA | 1.03e-06 (OR 0.95; 0.93–0.97) | NA |
| NS5B | 2986(R) | NA | NA | NA | NA | 2.9e-07 (OR 1.05; 1.03–1.07) | NA |
| NS5B | 2991(H) | NA | NA | NA | NA | 4.66e-12 (OR 0.88; 0.85–0.91) | NA |

*Table 2 continued on next page*

*Table 2 continued*

| HCV gene | Position (amino acid) | Genotype 1a N = 3548 | Genotype 1b N = 1924 | Genotype 2a N = 304 | Genotype 2b N = 472 | Genotype 3a N = 1839 | Genotype 4a N = 193 |
|---|---|---|---|---|---|---|---|
| NS5B | 2991(Y) | NA | NA | NA | NA | 1.86e-17 (OR 1.17; 1.13–1.22) | NA |
| NS5B | 3008(F) | 7.47e-08 (OR 1.01; 1.01–1.02) | NA | NA | NA | NA | NA |

DOI: https://doi.org/10.7554/eLife.42542.006

strongest associations that we observed were in and around this highly variable region, suggesting a possible role of these variants in determining the response to IFN-based antiviral treatment. The strongest association in the ISDR was with leucine at position 2224 in patients infected with 1b genotype, with higher mean viral load observed in presence of leucine for patients with the rs12979860 CC genotype. We also confirmed previously reported findings in the region, including associations with histidine at position 2065[18] (also known as the NS5A Y93H variant) and with asparagine at position 2414[11]. Using a genotype three replicon assay, Ansari et al. showed that this later variant - a change from a serine to asparagine at site 2414 - is associated with an increase in RNA replication, which is concordant with our results.

This is the first comprehensive analysis of IFN-λ-driven HCV adaptation across different viral genotypes and ancestry groups. In addition to identifying genotype or ancestry-specific associations, we observed sites of interaction that were consistent across HCV genotypes and ethnicities; for example, the NS5A variant Y2065H, which was found to be associated with rs12979860 in individuals infected with HCV genotypes 1a and 1b. These results indicate that IFN-λ-driven viral adaptation is a part of evolution across HCV genotypes.

In an attempt to delineate the biological impact of these associations, we evaluated the associations between HCV amino acid variants and pre-treatment viral load. We were able to detect a subset of amino acids that associated with both IFN-λ variation and HCV viral load across different viral genotypes, supporting the clinical relevance of host and pathogen interactions. Furthermore, we also performed a similar analysis with residual viral load, that is the fraction of the viral load variance that that is not explained by IFN-λ variation. We detected a group of viral amino acid variants that associated with SNP variations as well as residual viral load, indicating a stronger role of host–pathogen interactions in explaining the variations in HCV viral load.

Interestingly, only a fraction of the host-driven HCV amino acid variants was found to be associated with viral load, indicating that an integrated association analysis between host and pathogen genome variations can reveal correlations that would go unnoticed in association studies that use more downstream laboratory measurements or clinical outcomes as phenotypes.

IFN-λ polymorphism is the strongest human genetic predictor of spontaneous HCV clearance and response to IFN-based therapy. By integrating IFN-λ and HCV amino acid variation in a joint analysis, we here contribute to a better understanding of the genomic mechanisms involved in inter-individual differences in HCV disease outcomes. Our results confirm that IFN-λ4 is a functional gene that plays a pivotal role in HCV pathogenesis. The large footprint left by IFNL4 variation on the HCV proteome is indeed a clear indicator of the importance of innate immunity in viral control and of the remarkable capacity of HCV to evolve escape strategies.

## Materials and methods

### Clinical samples

Across 82 studies involving >100 sites in many countries, appropriate informed consent was obtained from study participants allowing the current analysis to be performed (*Welzel et al., 2017*). The studies were run by Gilead Sciences (Foster City, CA) and Pharmasset (formerly Princeton, NJ). Study protocols followed the ethical guidelines set in place by the 1975 Declaration of Helsinki and were approved by the relevant institutional review board committees. All samples included in this analysis are baseline samples collected from treatment naive and experienced patients

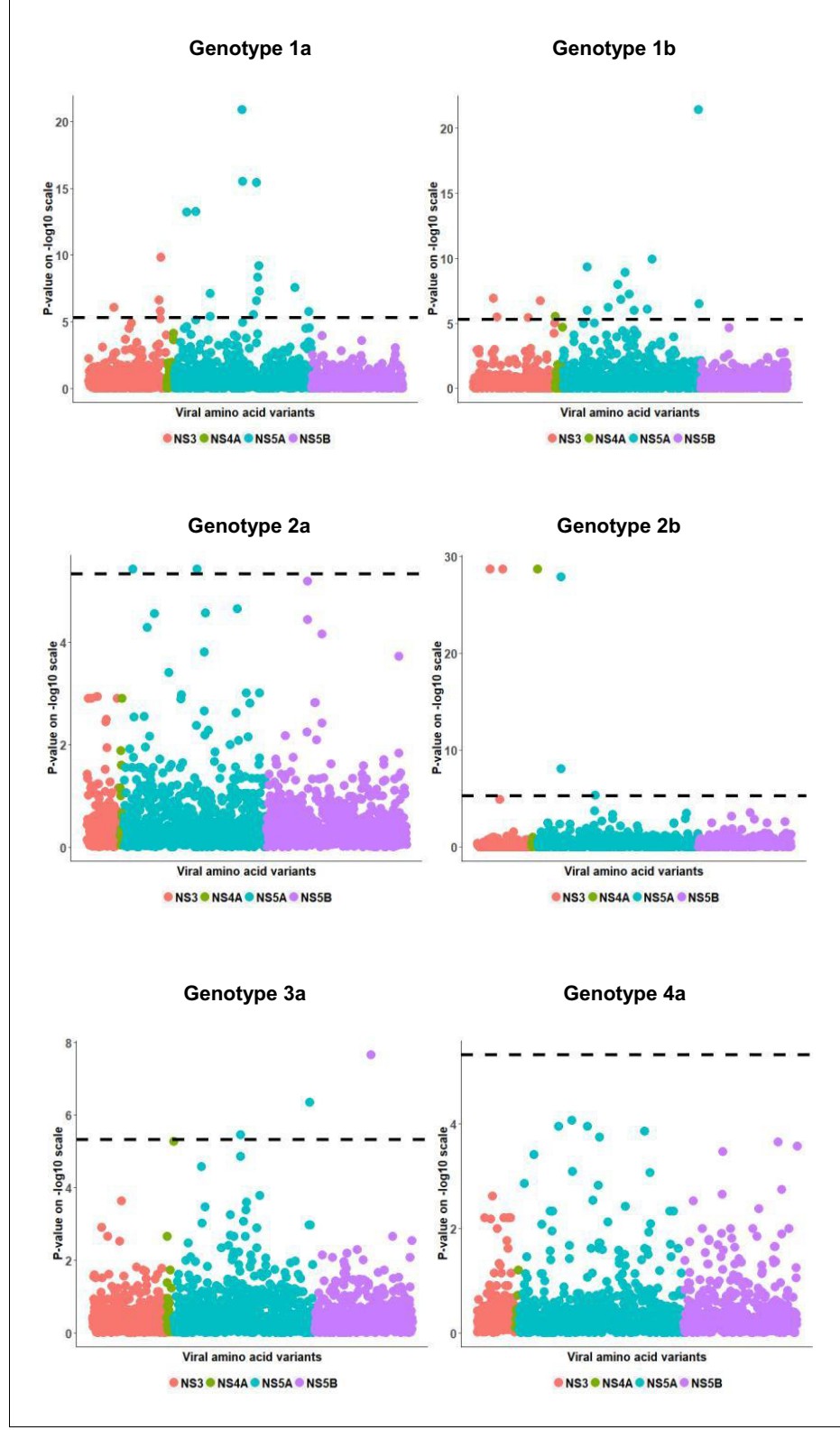

**Figure 2.** Per genotype viral load GWAS analysis results. Manhattan plot for associations between human Box-Cox transformed pre-treatment viral load and HCV amino acid variants. The dotted line shows the Bonferroni-corrected significance threshold.

DOI: https://doi.org/10.7554/eLife.42542.007

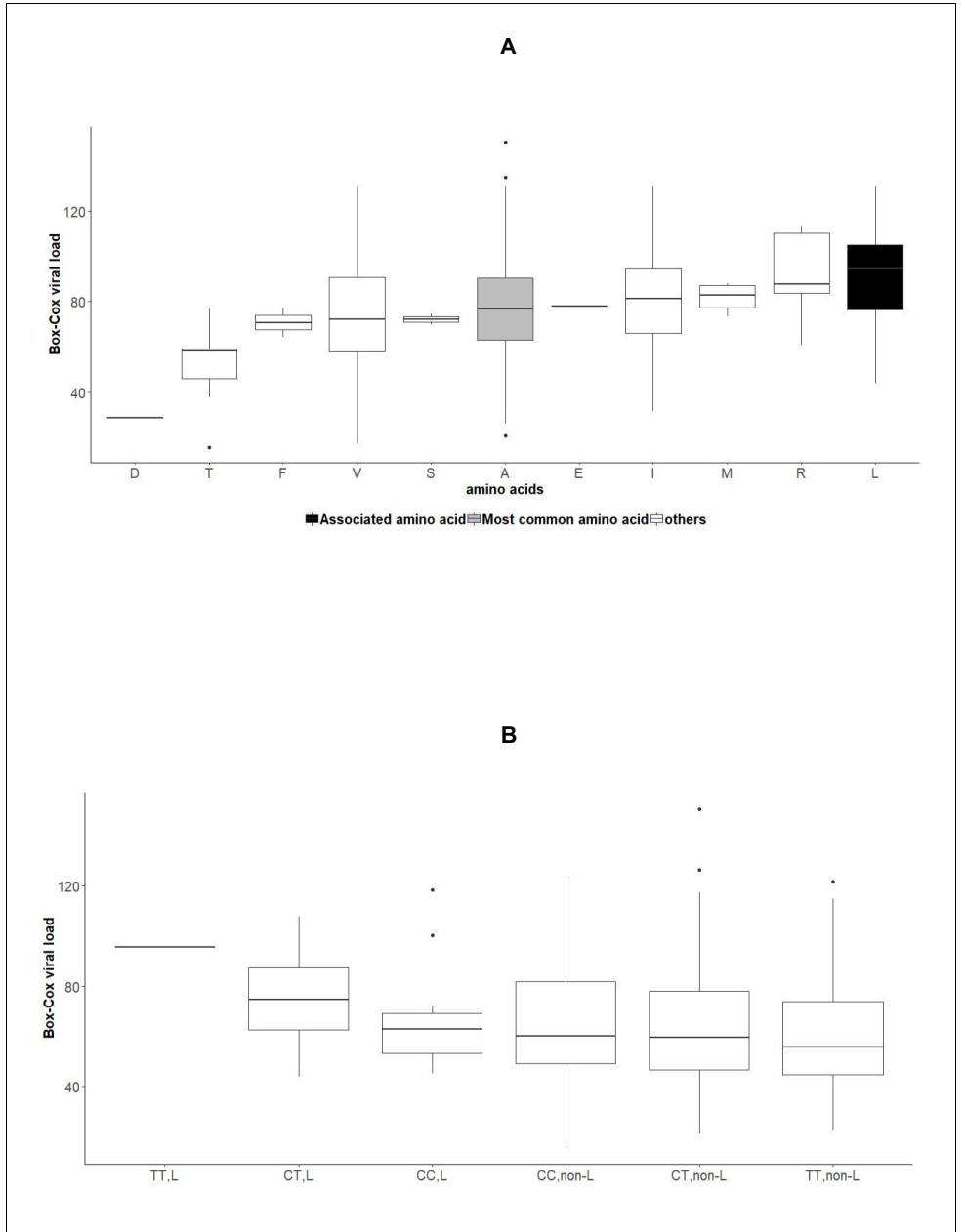

**Figure 3.** Associations between amino acid variables at position 2224 of NS5A, rs12979860 genotypes and HCV viral load in the group of patients infected with HCV genotype 1b. (A) Boxplot of transformed viral load stratified by amino acids present at position 2224 of NS5A. (B): Boxplot of transformed viral load stratified by rs12979860 genotypes (CC, CT, TT) and by presence or absence of leucine at position 2224 of NS5A.

DOI: https://doi.org/10.7554/eLife.42542.008

The following figure supplements are available for figure 3:

**Figure supplement 1.** Boxplot of transformed viral load stratified by rs12979860 genotypes (CC, CT, TT) in samples infected with viral genotype 3a, whose virus carries Serine at position 2414.

DOI: https://doi.org/10.7554/eLife.42542.009

**Figure supplement 2.** Per genotype viral load residual analysis results.

DOI: https://doi.org/10.7554/eLife.42542.010

**Figure supplement 3.** Per genotype integrated association analysis results in the European subgroup.

DOI: https://doi.org/10.7554/eLife.42542.011

**Figure supplement 4.** European per genotype viral load GWAS analysis results.

DOI: https://doi.org/10.7554/eLife.42542.012

*Figure 3 continued on next page*

*Figure 3 continued*

**Figure supplement 5.** European per genotype viral load residual GWAS analysis results.
DOI: https://doi.org/10.7554/eLife.42542.013

from >25 countries in North America, Europe, Asia, Oceania, and Africa between years 2010 and 2015.

## NS3, NS5A, and NS5B sequencing

The genotype assignment from Siemens VERSANT HCV Genotype INNO-LiPA 2.0 Assay (Innogenetics, Ghent, Belgium) was used to select genotype-specific primers located outside of the gene target(s) that amplify the entire *NS3/4A*, *NS5A*, or *NS5B* regions of HCV. Standard reverse transcription polymerase chain reaction (RT-PCR) was performed on patient plasma with HCV RNA >1000 IU/mL at DDL Diagnostic Laboratory (Rijswijk, The Netherlands). For deep sequencing, amplicons encoding the subject-derived *NS3/4A*, *NS5A* and *NS5B* were run using Illumina MiSeq v2 150 paired-end deep sequencing at DDL or WuXi AppTec (Shanghai, China). FASTQ files were split based on 100% matched barcodes. Contigs were generated from paired-end FASTQ files using VICUNA (*Yang et al., 2012*) and merged to create a de novo assembly sequence. All paired-end reads were merged using PEAR (*Zhang et al., 2014*), chopped at the 3' end when MAPQ <15, and filtered to remove reads <50 bases. The filtered reads were aligned to the de novo assembly sequence using MOSAIK (*Lee et al., 2014*) (v1.1.0017) to create a final assembly sequence. The average coverage of >5000 reads per position was obtained for most of the samples. The aligned reads were translated in-frame and the resulting tabulated summary of variants from the final assembly was utilized to generate a consensus sequence. Mixtures were reported when present in ≥15% of the viral population. NS3/4A, NS5A and NS5B consensus nucleotide and amino acid sequences were compared by the NCBI alignment tool BLAST to a set of reference sequences to assign HCV genotype and subtype. Amino acid variation between the samples that were assigned to genotype 1a, 1b, 2a, 2b, 3a and 4a were tabulated and analyzed. The raw HCV sequences are available in the zenodo repository, https://doi.org/10.5281/zenodo.1476713.

## Host genotyping

Human genotype was determined by PCR amplification and sequencing of the rs12979860 SNP region. Possible genotypes were CC, CT or TT.

## Association analyses

To run the integrated association analysis between genotyped host SNP and viral amino acids, we used logistic regression where the traits of interest were the presence or absence of each amino acid at the variable sites of the virus proteome. We assumed an additive model and corrected for host population stratification by adding sex, country of origin, self-reported ethnicity, cirrhosis status and prior treatment experience as covariates. To account for residual viral stratification within each HCV genotype, the first five phylogenetic principal components (*Revell, 2009*), calculated per HCV gene to account for recombination, were also added as covariates.

For the viral load GWAS analysis, we used linear regression where the trait of interest was Box-Cox transformed pre-treatment viral load. We used Box-Cox transformation to transform the positively skewed viral load distribution into a normally distributed dependent variable. We corrected for host and viral population stratification by adding sex, country of origin, self-reported ethnicity, cirrhosis status and prior treatment experience, as well as the first five viral phylogenetic principal components as covariates.

To correct for multiple testing we calculated the Bonferroni threshold as $\frac{0.05}{n^A}$, where $n^A$ represents the number of tests performed. For the analyses described in the paper, we performed a total of 10,681 tests. Given the heterogeneity of the dataset with multiple genotypes and ethnicities, we performed the integrated association analysis as well as viral load GWAS analyses on different sample subsets, created per genotype as well as per ethnic group.

## Software used

We used muscle (*Edgar, 2004*) to align the pathogen sequences, RaXML (*Stamatakis, 2014*) to obtain the phylogenetic trees and R (*R Development Core Team, 2013*) for all other analyses.

# Additional information

### Competing interests

Evguenia S Svarovskaia, Hongmei Mo, Anu O Osinusi, Diana M Brainard, G Mani Subramanian, John G McHutchison: This study was partially funded by Gilead Sciences and the author is an employee of Gilead Sciences. Stefan Zeuzem: has been a consultant for Abbvie, Gilead, Janssen, Merck/MSD. The other authors declare that no competing interests exist.

### Funding

| Funder | Grant reference number | Author |
|---|---|---|
| Gilead Sciences | | Jacques Fellay |
| Swiss National Science Foundation | PP00P3_157529 | Jacques Fellay |

The funders had no role in study design, data collection and interpretation, or the decision to submit the work for publication.

### Author contributions

Nimisha Chaturvedi, Conceptualization, Data curation, Formal analysis, Methodology, Writing—original draft, Writing—review and editing; Evguenia S Svarovskaia, Resources, Data curation, Writing—review and editing; Hongmei Mo, Anu O Osinusi, Diana M Brainard, G Mani Subramanian, John G McHutchison, Resources, Writing—review and editing; Stefan Zeuzem, Writing—review and editing; Jacques Fellay, Conceptualization, Funding acquisition, Writing—review and editing

### Author ORCIDs

Nimisha Chaturvedi (iD) https://orcid.org/0000-0002-3065-0202
Jacques Fellay (iD) https://orcid.org/0000-0002-8240-939X

### Ethics

Human subjects: Across 82 studies involving <100 sites in many countries, appropriate informed consent was obtained from study participants allowing the current analysis to be performed. The studies were run by Gilead Sciences (Foster City, CA) and Pharmasset (formerly Princeton, NJ). Study protocols followed the ethical guidelines set in place by the 1975 Declaration of Helsinki and were approved by the relevant institutional review board committees (further details for the studies can be found in Supplementary Table 1 in Welzel et al. [Journal of Hepatology, 2017]). All samples included in this analysis are baseline samples collected from treatment naive and experienced patients from <25 countries in North America, Europe, Asia, Oceania, and Africa between years 2010 and 2015.

### Decision letter and Author response

Decision letter https://doi.org/10.7554/eLife.42542.023
Author response https://doi.org/10.7554/eLife.42542.024

# Additional files

### Supplementary files

• Supplementary file 1. HCV amino acid positions with significant association p-values from genome-to-genome analysis (column 3), viral load GWAS analysis (column 4) and viral load residual GWAS

analysis (column 5) for viral genotype 1a. HCV genes and positions on the HCV proteome are given in the first and the second column of the table. Amino acid residuals on the associated positions are given in the second column.

DOI: https://doi.org/10.7554/eLife.42542.014

• Supplementary file 2. HCV amino acid positions with significant association p-values from genome-to-genome analysis (column 3), viral load GWAS analysis (column 4) and viral load residual GWAS analysis (column 5) for viral genotype 1b. HCV genes and positions on the HCV proteome are given in the first and the second column of the table. Amino acid residuals on the associated positions are given in the second column.

DOI: https://doi.org/10.7554/eLife.42542.015

• Supplementary file 3. Genome to genome analysis results for Asians and Europeans. The table consists of significant p-values and NA represents non-significant p-values.

DOI: https://doi.org/10.7554/eLife.42542.016

• Supplementary file 4. Genome to genome analysis results per genotype in European samples. The table consists of significant p-values and NA represents non-significant p-values.

DOI: https://doi.org/10.7554/eLife.42542.017

• Supplementary file 5. HCV amino acid positions with significant association p-values from genome-to-genome analysis (column 3), viral load GWAS analysis (column 4) and viral load residual GWAS analysis (column 5), for European samples infected with viral genotype 1a. HCV genes and positions on the HCV proteome are given in the first and the second column of the table. Amino acid residuals on the associated positions are given in the second column.

DOI: https://doi.org/10.7554/eLife.42542.018

• Transparent reporting form

DOI: https://doi.org/10.7554/eLife.42542.019

### Data availability

The raw HCV sequences are available in the Zenodo repository, https://doi.org/10.5281/zenodo.1476713. Patients did not explicitly consent to their data being made public and access to the human rs12979860 genotypes and relevant demographic and clinical variables is therefore restricted. Requests for the anonymized data should be made to Evguenia Svarovskaia (Evguenia.Svarovskaia@gilead.com) and will be reviewed by a data access committee, taking into account the research proposal and intended use of the data. Requestors are required to sign a data sharing agreement to ensure patients' confidentiality is maintained prior to the release of any data.

The following dataset was generated:

| Author(s) | Year | Dataset title | Dataset URL | Database and Identifier |
|---|---|---|---|---|
| Chaturvedi N, Svarovskaia ES, Mo H, Osinusi AO, Brainard DM, Subramanian GM, McHutchison JG, Zeuzem S, Fellay J | 2018 | Pervasive Adaptation Of Hepatitis C Virus To Interferon Lambda Polymorphism Across Multiple Genotypes | https://doi.org/10.5281/zenodo.1476713 | Zenodo, 10.5281/zenodo.1476713 |

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
