## [Decision Letter]

Thank you for submitting your article "Pervasive adaptation of hepatitis C virus to interferon lambda polymorphism across multiple genotypes" for consideration by *eLife*. Your article has been reviewed by four peer reviewers, one of whom is a member of our Board of Reviewing Editors, and the evaluation has been overseen by Wenhui Li as the Senior Editor. The reviewers have opted to remain anonymous.

The reviewers have discussed the reviews with one another and the Reviewing Editor has drafted this decision to help you prepare a revised submission.

Summary:

Two papers by Ansari et al. are highly relevant to this submission. Previously, Ansari et al. reported results of a genome-to-genome study of 542 individuals who were chronically infected with HCV viral genotype (VGT) 3 (predominantly) or VGT 2. (Nature Genetics, 2017) In a joint submission with the present paper, Ansari et al. now report analyses restricted to the subset of those subjects who were infected with HCV genotype 3a. Results from both of those papers should be referenced in the present paper, as appropriate.

The present paper examines associations between genotype for IFNL4 rs12979860 (a marker for rs368234815) and variation in four HCV proteins (NS3, NS4A, NS5A and NS5B) among 8,729 HCV-infected patients enrolled in clinical trials conducted by Gilead Sciences. These individuals were infected with a range of VGTs (1a – 3,548; 1b – 1,924; 2a- 304; 2b- 472; 3a- 1,839; 4a-193). Thus, the authors have data to replicate findings of Ansari et al. regarding VGT3, examine associations with other VGTs and compare viral associations with IFNL4 genotype across VGTs. Given the richness of this resource, the analysis is disappointing; the paper needs substantial additional analyses and a new set of figures. In addition, the study population and the statistical methods must be described more clearly. With significant revisions, this paper could make an important contribution to the HCV literature.

Essential revisions:

The lack of line numbers or page numbers makes it difficult to convey comments. Please add line numbers (or at least page numbers) to the manuscript.

Study population and design:

Given that baseline samples are analyzed in relation to baseline outcome measures, this is a cross sectional study rather than a cohort analysis. The terminology should be clarified, as otherwise it is misleading.

The authors only genotype the SNP rs12979860 and, therefore, cannot distinguish between the two haplotypes encoding the IFNL4 P70 and IFNL4 S70 proteins per the co-submission. This limitation should be acknowledged in the Discussion.

Nomenclature:

This paper uses protein-specific numbering of amino acid positions, whereas Ansari et al. employ virome wide numbering. For the sake of comparison, the present paper should include virome wide numbering.

"a genetic polymorphism in the interferon region" is ambiguous as there are multiple polymorphisms in this region. "rs12979860 genotype, a tag for IFN-λ haplotypes" or "rs12979860, a known marker of IFN λ haplotypes" is also ambiguous and not relevant as haplotypes are not explained or used in this paper. The most relevant point re: rs12979860 that it is a marker for rs368234815.

Statistical analyses and results:

Consensus viral sequences and the propensity for specific viral variants vary markedly by VGT, therefore, associations between IFNL4 genotype and viral variants vary by VGT. This is demonstrated in Supplementary Table 3, which provides key data. A revised version of this table (or a derived figure) should be included in the body of the paper. The organizing principle for presentation of the HCV variants in Supplementary Table 3 is unclear. The table should be simplified by restricting it to results of either additive or recessive genetic models, based on an evaluation of which model is more appropriate.

Given these observed differences in viral associations in different VGTs, it is inappropriate to combine all 8,729 patients in analyses. These subjects comprise a convenience sample of patients enrolled in the clinical trials, which is not a population of intrinsic biological interest. (A different set of trials amongst patients with a different distribution of VGTs would yield different summary data.) Figures 1, 2B and 3 should be deleted; Figures 1 and 3 should be replaced by sets of VGT-specific figures.

Table 1 provides little information. The paper should include a table that describes the characteristics of the subjects (e.g., sex, age, race, country of enrollment and HCV disease status), by VGT.

The presentation of results of the statistical analysis is difficult to follow. The authors should specify the associations. For example, the presence of a functional IFNL4 variant associates with a proline at position XXX in viral protein Y. The data presented on the correlation of specific HCV amino acids is interesting but equally underdeveloped.

The authors largely present results as p-values, however, to evaluate the effect size, the amplitude of the signals should be presented as well.

The data describing HCV viral load should be analyzed in more detail and presented more clearly. It should be possible for the reader to determine if a given amino acid change associates with higher or lower viral load, how higher or lower viral load associates with the three major IFNL genotypes, and the effect of viral load of being heterozygous.

Describe the effect of being heterozygous more clearly to the reader, often two correlation coefficients are listed (recessive and additive) but without explanation and only the p-value is listed. The viral load data should be analyzed more carefully, including using unadjusted and adjusted regression models, to inform the reader about what the effect of this particular genotype is on viral load.

In association of host and viral genotype, how was the Bonferroni threshold determined?

In the logistic models used to assess associations, what was the outcome? Host or viral genotype? Were the models adjusted for any covariates? Please specify.

Were models considered that contained more than one viral amino acid, i.e. were viral results mutually adjusted to better pinpoint where the signal is coming from?

The association of IFNL4 genotype with the frequency of HCV polymorphisms could reflect an effect of IFNL4 on viral replication rates. To assess that possibility, the investigators should compare the results of two logistic regression models: one that does and one that does not include HCV RNA as an additional covariate to IFNL4. Otherwise these paired models should include identical adjustments.

Viral load is a quantitative trait and it is important that proper methods be used in those analyses. Residual regression analysis (subsection “HCV amino acid variants and viral load”): what are the assumptions made here, e.g. normal distribution of viral load? Was viral load transformed?

Supplementary Table 6 – There is no mention of VL GWAS in the main text or Materials and methods. Please explain this approach in Materials and methods and table footnotes; comment on differences in p-values between VL GWAS and G2G analysis.

Discussion:

The results should be discussed in the context of current knowledge from the literature regarding the function of rs368234815. HCV does not adapt to a genetic polymorphism, but to its functional effect, which in this case is likely the production of IFN-λ4.

Other:

Throughout the paper, clearly state what is a new finding in this paper and what confirms or contradicts the two papers by Ansari et al.

Supplementary tables should include footnotes explaining the analyses with regard to the statistical model employed, adjustments and significance thresholds. Also, the number of subjects in each subgroup.

Re: title, suggest: Adaptation of hepatitis C virus to interferon lambda polymorphism across multiple viral genotypes.

Additional data files:

Current data sharing covers only HCV variation, not the human rs12979860 genotypes and relevant demographic and clinical variables. It would be important to make this unique dataset available to the scientific community through controlled dbGAP access. Otherwise, this analysis cannot be validated.

[Editors' note: further revisions were requested prior to acceptance, as described below.]

Thank you for resubmitting your work entitled "Adaptation of hepatitis C virus to interferon lambda polymorphism across multiple viral genotypes" for further consideration at *eLife*. Your revised article has been favorably evaluated by Wendy Garrett as the Senior Editor and a Reviewing Editor.

The manuscript has been improved but there are some remaining issues that need to be addressed before acceptance, as outlined below:

Summary:

The authors were very responsive to the reviewer comments. The revised paper, which includes substantial additional analyses and a new set of figures, is much improved. The new analyses demonstrate that most of the associations between IFNL4 genotype and viral variants are viral genotype (VGT) specific, as would be expected given that consensus viral sequences and the propensity for specific viral variants vary markedly by VGT. This paper will be an important contribution to the HCV literature.

Essential revisions:

Previous comments requiring further attention.

1) In association of host and viral genotype, how was the Bonferroni threshold determined?

More details on the regression model as well as significance threshold are given in the Materials and methods section (subsection “Association analyses”).

Unfortunately, that information is not enough. Specify the number of tests performed. If that number varies meaningfully for the different analyses, then the number should be specified each time the Bonferroni correction is used.

2) Viral load is a quantitative trait and it is important that proper methods be used in those analyses. Residual regression analysis (subsection “HCV amino acid variants and viral load”): what are the assumptions made here, e.g. normal distribution of viral load? Was viral load transformed?

We did use Box-Cox transformation for viral load and then used the viral load residuals obtained from transformed viral load. We have added more information on this is in the Materials and methods section (subsection “Association analyses”).

The description in the subsection “Association analyses” states "we used logistic regression where the trait of interest was Box-Cox transformed pre-treatment viral load." However, logistic regression outcomes are usually binary variables. This statement,"Viral load GWAS analysis was performed using linear regression, between transformed viral load as trait of interest and viral amino acid variations." Please clarify what was done.

New comments arising from the extensive revisions.

1) The Abstract should be more specific. Describe the distribution of viral genotypes examined and some key findings.

2) The term 'genome-to-genome' seems inappropriate for these analyses. 'Genome' implies a complete set of genetic information; however, these analyses involve a single human variant and variants from a selected set of HCV proteins.

3) Subsection “Viral load association analyses”, second paragraph – I found this text confusing. Suggest the following revision (if correct): 'We then searched for associations between viral load and HCV amino acid variables. These analyses identified significant associations in all viral genotype groups except 4a (Figure 2). Amongst the viral amino acids that associated with viral load, a number also associated with rs12979860 genotype (genotype 1a, 9 of x amino acids; 1b, 5 of x amino acids; 2a, 0 of x amino acids; 2b, 0 of x amino acids; 3a, 2 of x amino acids).'

4) Subsection “Viral load association analyses”, third paragraph – Please clarify if this analysis is restricted to individuals infected with genotype 1b. Please verify that Figure 3B is consistent with this text and that the y axis is not mislabeled.

5) Table 2 presents '97% CI', rather than 95% CI, which are more usual. If 97% is in fact correct, provide a justification for using an interval of that width.

---

## [Author Response]

Essential revisions:The lack of line numbers or page numbers makes it difficult to convey comments. Please add line numbers (or at least page numbers) to the manuscript.

We have added the page and line numbers in the revised version.

Study population and design:Given that baseline samples are analyzed in relation to baseline outcome measures, this is a cross sectional study rather than a cohort analysis. The terminology should be clarified, as otherwise it is misleading.We have changed the terminology in the revised version, as suggested by the reviewer. Changes are in the last paragraph of the Introduction.The authors only genotype the SNP rs12979860 and, therefore, cannot distinguish between the two haplotypes encoding the IFNL4 P70 and IFNL4 S70 proteins per the co-submission. This limitation should be acknowledged in the Discussion.

We have added a part about this limitation in the Discussion (second paragraph).

Nomenclature:This paper uses protein-specific numbering of amino acid positions, whereas Ansari et al. employ virome wide numbering. For the sake of comparison, the present paper should include virome wide numbering.

We have changed the protein specific numbering of amino acids to virome wide numbering as suggested by the reviewer.

"a genetic polymorphism in the interferon region" is ambiguous as there are multiple polymorphisms in this region. "rs12979860 genotype, a tag for IFN-λ haplotypes" or "rs12979860, a known marker of IFN λ haplotypes" is also ambiguous and not relevant as haplotypes are not explained or used in this paper. The most relevant point re: rs12979860 that it is a marker for rs368234815.

We have incorporated the suggested changes in the paper and removed the mentions of rs12979860 genotype being a tag for IFN-λ haplotypes.

Statistical analyses and results:Consensus viral sequences and the propensity for specific viral variants vary markedly by VGT, therefore, associations between IFNL4 genotype and viral variants vary by VGT. This is demonstrated in Supplementary Table 3, which provides key data. A revised version of this table (or a derived figure) should be included in the body of the paper. The organizing principle for presentation of the HCV variants in Supplementary Table 3 is unclear. The table should be simplified by restricting it to results of either additive or recessive genetic models, based on an evaluation of which model is more appropriate.

We have moved the supplementary table in the main paper (Table 2) and added more details on VGT specific analysis in the Results section. Table 2 (previously Supplementary Table 3) now contains association results from the additive model only.

Given these observed differences in viral associations in different VGTs, it is inappropriate to combine all 8,729 patients in analyses. These subjects comprise a convenience sample of patients enrolled in the clinical trials, which is not a population of intrinsic biological interest. (A different set of trials amongst patients with a different distribution of VGTs would yield different summary data.) Figures 1, 2B and 3 should be deleted; Figures 1 and 3 should be replaced by sets of VGT-specific figures.

We have removed the results from the global analysis of our study population (including all figures and tables). The Results section now focuses on VGT specific analysis. Ancestry specific sub-analyses are mentioned in the Results section.

Table 1 provides little information. The paper should include a table that describes the characteristics of the subjects (e.g., sex, age, race, country of enrollment and HCV disease status), by VGT.

We have added more information in Table 1, per VGT, including ethnicity, sex, cirrhosis status and SVR status.

The presentation of results of the statistical analysis is difficult to follow. The authors should specify the associations. For example, the presence of a functional IFNL4 variant associates with a proline at position XXX in viral protein Y. The data presented on the correlation of specific HCV amino acids is interesting but equally underdeveloped.

We have changed the presentation of the results and incorporated the suggested modifications. We have also performed additional analyses to dissect specific association results and added the results to the revised version.

The authors largely present results as p-values, however, to evaluate the effect size, the amplitude of the signals should be presented as well.

We have added odds ratios and confidence intervals to the p-values presented in Table 2 (main results) as well as in the supplementary files that present analysis results.

The data describing HCV viral load should be analyzed in more detail and presented more clearly. It should be possible for the reader to determine if a given amino acid change associates with higher or lower viral load, how higher or lower viral load associates with the three major IFNL genotypes, and the effect of viral load of being heterozygous.

We have added more analysis results showing the correlation between HCV amino acid changes and viral load as well as host genotype and viral load (subsection “Viral load association analyses”).

Describe the effect of being heterozygous more clearly to the reader, often two correlation coefficients are listed (recessive and additive) but without explanation and only the p-value is listed. The viral load data should be analyzed more carefully, including using unadjusted and adjusted regression models, to inform the reader about what the effect of this particular genotype is on viral load.

We have now removed the results from the recessive analysis from the paper and provided more details on additive regression model in the Materials and methods section (subsection “Association analyses”).

In association of host and viral genotype, how was the Bonferroni threshold determined?

More details on the regression model as well as significance threshold are given in the Materials and methods section (subsection “Association analyses”).

In the logistic models used to assess associations, what was the outcome? Host or viral genotype? Were the models adjusted for any covariates? Please specify.

For the genome-to-genome analysis, we used a logistic regression based association model, where the binary HCV amino acids were the outcomes. We added multiple host as well as pathogen population covariates to account for population stratification and thus reduce the risk of obtaining false positives association results. We have added more details on the regression analyses in the Materials and methods section (subsection “Association analyses”).

The association of IFNL4 genotype with the frequency of HCV polymorphisms could reflect an effect of IFNL4 on viral replication rates. To assess that possibility, the investigators should compare the results of two logistic regression models: one that does and one that does not include HCV RNA as an additional covariate to IFNL4. Otherwise these paired models should include identical adjustments.

As suggested by the reviewer, we ran two series of logistic regression models, one with HCV RNA as an additional covariate and one without. The association results from the two models were very similar. We have added the results of these analyses in the last paragraph of the subsection “Genome-to-genome analyses” (Figure 1—figure supplement 1).

Viral load is a quantitative trait and it is important that proper methods be used in those analyses. Residual regression analysis (subsection “HCV amino acid variants and viral load”): what are the assumptions made here, e.g. normal distribution of viral load? Was viral load transformed?

We did use Box-Cox transformation for viral load and then used the viral load residuals obtained from transformed viral load. We have added more information on this is in the Materials and methods section (subsection “Association analyses”).

Supplementary Table 6 – There is no mention of VL GWAS in the main text or Materials and methods. Please explain this approach in Materials and methods and table footnotes; comment on differences in p-values between VL GWAS and G2G analysis.

We have added more information on the viral load GWAS (VL GWAS) in the Materials and methods subsection “Association analyses”.

Discussion:The results should be discussed in the context of current knowledge from the literature regarding the function of rs368234815. HCV does not adapt to a genetic polymorphism, but to its functional effect, which in this case is likely the production of IFN-λ4.

We have modified the Introduction and the Discussion accordingly.

Other:Throughout the paper, clearly state what is a new finding in this paper and what confirms or contradicts the two papers by Ansari et al.

We have added more comparisons between the results from our analysis and the results from Ansari et al.’s papers (subsection “Genome-to-genome analyses”, third paragraph and subsection “Viral load association analyses”, fourth paragraph), including replications as well as new results from our analysis that were not detected in the analysis by Ansari et al.

Supplementary tables should include footnotes explaining the analyses with regard to the statistical model employed, adjustments and significance thresholds. Also, the number of subjects in each subgroup.

We added this information in the supplementary files.

Re: title, suggest: Adaptation of hepatitis C virus to interferon lambda polymorphism across multiple viral genotypes.

We have changed the title to the one suggested by the reviewer.

Additional data files:Current data sharing covers only HCV variation, not the human rs12979860 genotypes and relevant demographic and clinical variables. It would be important to make this unique dataset available to the scientific community through controlled dbGAP access. Otherwise, this analysis cannot be validated.

Unfortunately, we are not able to provide individual-level demographic and clinical data. The consent form signed by study participants does not allow for the sharing of that information.

[Editors' note: further revisions were requested prior to acceptance, as described below.]

The manuscript has been improved but there are some remaining issues that need to be addressed before acceptance, as outlined below:Summary:The authors were very responsive to the reviewer comments. The revised paper, which includes substantial additional analyses and a new set of figures, is much improved. The new analyses demonstrate that most of the associations between IFNL4 genotype and viral variants are viral genotype (VGT) specific, as would be expected given that consensus viral sequences and the propensity for specific viral variants vary markedly by VGT. This paper will be an important contribution to the HCV literature.Essential revisions:Previous comments requiring further attention.1) In association of host and viral genotype, how was the Bonferroni threshold determined?More details on the regression model as well as significance threshold are given in the Materials and methods section (subsection “Association analyses”).Unfortunately, that information is not enough. Specify the number of tests performed. If that number varies meaningfully for the different analyses, then the number should be specified each time the Bonferroni correction is used.

We have now added the number of tests performed in the Materials and methods section (subsection “Association analyses”, last paragraph).

2) Viral load is a quantitative trait and it is important that proper methods be used in those analyses. Residual regression analysis (subsection “HCV amino acid variants and viral load”): what are the assumptions made here, e.g. normal distribution of viral load? Was viral load transformed?We did use Box-Cox transformation for viral load and then used the viral load residuals obtained from transformed viral load. We have added more information on this is in the Materials and methods section (subsection “Association analyses”).The description in the subsection “Association analyses” states "we used logistic regression where the trait of interest was Box-Cox transformed pre-treatment viral load." However, logistic regression outcomes are usually binary variables. This statement,"Viral load GWAS analysis was performed using linear regression, between transformed viral load as trait of interest and viral amino acid variations." Please clarify what was done.

We have fixed the mistake in the second paragraph of the subsection “Association analyses”, and replaced logistic regression with linear regression.

New comments arising from the extensive revisions.1) The Abstract should be more specific. Describe the distribution of viral genotypes examined and some key findings.

We have added more information in the Abstract.

2) The term 'genome-to-genome' seems inappropriate for these analyses. 'Genome' implies a complete set of genetic information; however, these analyses involve a single human variant and variants from a selected set of HCV proteins.

We have replaced the term “genome-to-genome analysis” with “integrated association analysis” throughout the article.

3) Subsection “Viral load association analyses”, second paragraph – I found this text confusing. Suggest the following revision (if correct): 'We then searched for associations between viral load and HCV amino acid variables. These analyses identified significant associations in all viral genotype groups except 4a (Figure 2). Amongst the viral amino acids that associated with viral load, a number also associated with rs12979860 genotype (genotype 1a, 9 of x amino acids; 1b, 5 of x amino acids; 2a, 0 of x amino acids; 2b, 0 of x amino acids; 3a, 2 of x amino acids).'

We have replaced the previous text with the one suggested by the reviewer (subsection “Viral load association analyses”, second paragraph).

4) Subsection “Viral load association analyses”, third paragraph – Please clarify if this analysis is restricted to individuals infected with genotype 1b. Please verify that Figure 3B is consistent with this text and that the y axis is not mislabeled.

Done (subsection “Viral load association analyses”, third paragraph). We checked the Figure 3B. It is consistent with the text and the y axis is not mislabeled.

5) Table 2 presents '97% CI', rather than 95% CI, which are more usual. If 97% is in fact correct, provide a justification for using an interval of that width.

Both 95% as well as 97% CI are correct. Since we had a large data size we could choose 97% CI to have more confidence in our results, without making the width between the lower and the upper bound too large. It can be seen from Table 2 that in spite of having 97% CI we have a narrow CI for the OR values and the lower and the upper range values of the CI are very close to the estimated OR values.